# Phloroglucinol Derivative Carbomer Hydrogel Accelerates MRSA-Infected Wounds’ Healing

**DOI:** 10.3390/ijms23158682

**Published:** 2022-08-04

**Authors:** Xiaosu Huang, Junhua Yang, Renyue Zhang, Lianbao Ye, Ming Li, Weiqiang Chen

**Affiliations:** 1School of Nursing, Guangdong Pharmaceutical University, Guangzhou 510006, China; 2Department of Anatomy, School of Biosciences & Biopharmaceutics, Guangdong Pharmaceutical University, Guangzhou 510006, China; 3School of Pharmacy, Guangdong Pharmaceutical University, Guangzhou 510006, China; 4School of Life Science and Biopharmaceutics, Guangdong Pharmaceutical University, Guangzhou 510006, China; 5Guangdong Key Laboratory of Pharmaceutical Bioactive Substances, Guangdong Pharmaceutical University, Guangzhou 510006, China

**Keywords:** phloroglucinol derivative carbomer hydrogel, MRSA infection, antibacterial, wound healing

## Abstract

Globally, wound infection is considered to be one of the major healthcare problems, with bacterial infections being the most critical threat, leading to poor and delayed wound healing, and even death. As a superbug, methicillin-resistant *Staphylococcus aureus* (MRSA) causes a profound hazard to public health safety, prompting us to search for alternative treatment approaches. Herein, the MTT test and Hoechst/propidium iodide (PI) staining demonstrated that PD was slightly less toxic to human fibroblasts including Human keratinocytes (HaCaT) cell line than Silver sulfadiazine (SSD), and Vancomycin (Van). In the MRSA-infected wound model, PD hydrogel (1%, 2.5%) was applied with for 14 days. The wound healing of PD hydrogel groups was superior to the SSD, Van, and control groups. Remarkably, the experimental results showed that PD reduced the number of skin bacteria, reduced inflammation, and upregulated the expression of PCNA (keratinocyte proliferation marker) and CD31 (angiogenesis manufacturer) at the wound site by histology (including hematoxylin–eosin (HE) staining, Masson staining) and immunohistochemistry. Additionally, no toxicity, hemocompatibility or histopathological changes to organs were observed. Altogether, these results suggested the potential of PD hydrogel as a safe, effective, and low toxicity hydrogel for the future clinical treatment of MRSA-infected wounds.

## 1. Introduction

Bacterially infected wounds can lead to delayed healing, tissue damage, and even the involvement of organs or death. Although the advances in the development of silver-based dressings, coatings, nanofibers, and hydrogels, bacterial wound infections remain a challenge worldwide [1]. Nevertheless, an optimal wound protection product to solve the problems of chronic wounds, including infection control, regulation of inflammation, and promotion of tissue regeneration, is still lacking. More noteworthy, the increasing incidence of acute and chronic wounds, coupled with alarming rates of antimicrobial resistance, and ineffective antibiotics, suggests the urgent requirement for alternative antimicrobial agents, a challenge that could otherwise escalate to the scale of a global epidemic [2].

Particularly, MRSA is one of the world’s most notoriously antibiotic-resistant members. MRSA infection causes a tremendous challenge to clinical care on human health and the healthcare system including higher mortality rates, longer hospital stays, and high medical costs. The WHO has classified MRSA as a high-priority target for new antibiotic development with the reason that it causes systemic infections including skin and soft tissue infections (SSTIs), septic arthritis, bacteremia, sepsis, infective endocarditis, and hospital-acquired infections, which seriously affect the health of humans and other organisms with high morbidity and mortality [3]. Notably, bacterial evolution has greatly outpaced the development of antimicrobial drugs. Hitherto, vancomycin has been considered the last line of defence against MRSA; however, the emergence of vancomycin-resistant bacteria has been reported [4]. Furthermore, antibiotics for MRSA also demonstrate non-specificity, as they kill useful human bacteria, leading to dysbiosis (microbial imbalance). Another tremendous challenge is the evolution of bacteria due to the abuse of antibiotics. It was reported that some MRSA strains have exhibited drug resistance to newer antibiotics such as daptomycin and last-resort antibiotics such as vancomycin and linezolid [5]. Of note, MRSA is one of the most difficult bacterial pathogens to treat; infections with antibiotic-resistant bacteria kill 1.8 million people each year, more than AIDS, tuberculosis, and viral hepatitis. By 2050, it is predicted that the number of deaths will increase dramatically to 10 million annually [6].

The Gram-positive bacterium *Staphylococcus aureus* is a commensal bacterium that is ubiquitous in human skin. Skin wound infections, particularly caused by drug-resistant pathogens, are a global problem, costing millions of dollars per year to treat, commonly leading to difficult wound healing and other infection-related severe syndromes such as bacteremia, lethal sepsis, acute renal failure, and so on, and still has a mortality rate of around 30% worldwide [7]. The skin is the body’s greatest organ and is the basis for maintaining the skin barrier, which is considered to be the basis for protection against pathogenic bacteria and other microbial infections.

The US Food and Drug Administration (FDA) has encouraged the use of effective combination therapies, involving vancomycin with β-lactam, daptomycin with β-lactams, rifampin with vancomycin, or daptomycin and other antibiotics [8], which has presented positive results. However, antibiotic combinations for MRSA have more serious adverse effects, as illustrated by a randomized clinical trial that showed a higher incidence of nephrotoxicity and toxicity with vancomycin in combination with β-lactams. Wound healing follows spontaneously through three successive phases: inflammation, proliferation, and remodeling [9]. However, when a wound is infected, the healing process is delayed during the inflammatory phase, causing damage to the wound. Therefore, eradication of bacteria from the injured site is an important step in the healing of infected wounds. With these considerations in mind, those situations presented a daunting challenge to treat MRSA infections.

Silver Sulfadiazine (SSD) is an effective antibacterial agent [10]. SSD 1% cream has good bacterial activity against most Gram-positive and Gram-negative bacteria. Currently, SSD is marketed as a cream/binder/powder/gel at a concentration of 1% *w*/*w*. In addition, propylene glycol, which is part of the SSD formulation, is known to cause bone marrow toxicity and leukopenia [11]. Other limitations of SSD include the need to change the dressing frequently, at least once a day, as it can lose its silver ions early and cause fear and pain, especially in children. Antibiotics, for which vancomycin is often used, are loaded into the hydrogel to prevent infection at the wound site. Additionally, although vancomycin is an ideal drug for the treatment of MRSA infections, it has some disadvantages including poor penetration, slow bactericidal effect, and serious damage to the ear and kidney which limits its clinical application [12]. However, antibiotic development may further stimulate bacterial evolution and thus the development of drug resistance.

The colonization of a wound site by pathogenic bacteria is a major cause of delayed wound healing. Therefore, preventing the growth of pathogenic bacteria in the wounds helps to promote wound healing. In some studies, phloroglucinol was found to exhibit anticancer [13], anti-inflammatory [14], and antibacterial activities [15]. Due to their potential antibacterial activity, some phloroglucinol has been synthesized, and its structure–activity relationships and mechanisms of action have also been studied further [16]. It was reported that phlorotannins possesses biological activities including antimicrobial, antidiabetic, anticancer, antiviral, and antioxidant [17]. Phloroglucinol encapsulated in chitosan nanoparticles by ionic gel technology is more effective against bacteria such as *Staphylococcus aureus* than pure phloroglucinol [18]. The addition of the polyphenol phloroglucinol to chitosan hydrogels reduced the growth of *E. coli* and also improved the antioxidant activity of the hydrogels [19]. Several studies have discussed the promising future of phloroglucinol and its derivatives in conjugation, nanoformulations, antibiotic combinations and encapsulation in antibacterial applications [20]. All over the world, governments and health authorities must seek ways to solve the intractable problem of antimicrobial resistance.

Given the critical situation caused by drug-resistant bacteria and the many reports finding good antibacterial effects of phloroglucinol, there is a goal of developing new treatments and reducing the medical burden associated with MRSA-infected skin wounds. Therefore, we synthesised PD and we aimed to investigate whether it has low toxicity to cell growth in vitro and better anti-MRSA wound infection in vivo.

## 2. Results

### 2.1. Phloroglucinol Derivatives Are Less Toxic to Human Keratinocytes (HaCaT) Than SSD and Van

As is presented in Figure 1A, the cell viability of HaCaT cells treated with 250 µg/mL of PD was 96.69 ± 5.67%, and that of SSD and Van wer 84.75 ± 17.35 and 96.34 ± 9.34%, respectively. Notably, at 250 µg/mL of SSD and Van, most of the cells died and floated, with cell survival rates of only 32.56 ± 4.05 and 40.68 ± 6.34%, respectively, as compared to 54.34 ± 5.67% under PD treatment. Detection of apoptosis by Hochest/PI staining of cells treated with 125 µg/mL for 24 h showed similar rates of apoptosis for SSD and Van, and lower rates for PD when compared to SSD and Van (Figure 1B). These findings suggested that PD is less toxic in human fibroblast cells compared to SSD and Van at the same concentration.

### 2.2. PD Effectively Controls MRSA Wound Infection and Promotes Wound Healing

We further examined the bacteria inhibition and wound-healing capacity of our PD hydrogel using an MRSA-infected wound model (Figure 2A). Comparisons between the control and blank group were both statistically significant, indicating that our model was successful. Quantitative measurements of the cross-sectional length of the histological wound also substantiated the finding, with the unhealed edge length in the PD group being (815.4 ± 146.7 µm), SSD (1751 ± 156.3 µm), Van (3433 ± 166.3 µm), and the control group (4306 ± 109.3 µm) (*p* < 0.05) (Figure 2C). As seen in Figure 2B,D, which shows the macroscopic picture of the wound and the percentage of the wound area, it is clear to see that PD can significantly promote wound healing compared to other treatments. According to the digital photographs and schematic graph of wound size (Figure 2B,C), the wound size shrank faster in the PD and SSD groups than in the control group. Simultaneously, as indicated in Figure 2D, the wound size graphs for each experimental group demonstrated significant changes on days 0, 3, 7, 11, and 14. The average wound size after PD or SSD treatment was 12.79 and 20.32%, respectively, smaller than the control group (53.29%) (*p* < 0.05). In addition, as presented in Figure 2E, on day 14, the mean wound healing areas for the low and high concentrations of PD hydrogels treatment were approximately 91.26 and 96.56%, respectively, while the mean wound healing area of the SSD and Van was approximately 89.45 and 87.77%. Compared to the control, SSD, and Van group, the 2.5% PD gel was statistically significantly different (*p* < 0.05). However, there was no significant statistical difference in the healing of the wound between SSD and Van groups (*p* > 0.05).

### 2.3. Colony Counting

Wound recovery is a complex process involving collagen synthesis, epithelial regeneration, synthetic tissue, and remodeling of the extracellular matrix. As depicted in Figure 3A, MRSA bacteria were incubated on agar plates for 24 h. Although the bacterial counts on days 3, 7, and 11 were not statistically significant, we could clearly see that the infected skin wounds of the mice improved after treatment (Figure 3). There was a significant reduction in the number of bacteria in the wounds in all treatment groups on day 14 compared to the control group, and the 2.5% PD gel group had fewer bacteria than both SSD and Van groups (*p* < 0.05) (Figure 3B).

### 2.4. Histopathological Studies and Collagen Deposition in Skin Tissue

The results of H&E staining showed that almost no scarring was observed in all PD-treated mice on the 14th day. PD did not present inflammation or oedema. Mice in the blank group formed an intact epidermis (Figure 4A). However, the intradermal distance was large, and inadequate keratinization, epidermal hyperplasia, and inflammatory infiltration were still present in the control group, indicating incomplete wound healing. Further, the epithelial layer in the PD group had a thicker epithelial layer than the other groups. The PD group indicated the epithelium at the incision had formed and the basal layer, spine layer, granular layer, hyaline layer, and a small amount of stratum corneum were visible. Of these dosing groups, the PD group had the most mature epidermal layer. However, in the SSD and Van groups, there were still areas that failed to have a complete epidermis. Masson staining showed that on day 14, blue collagen fibres appeared in all groups, including the blank and dosed groups (Figure 4B), and by day 14, an epidermal layer resembling a normal skin structure had formed. In the control group, this process was less active. (Figure 4B) The epithelial gap was significantly smaller in the PD group compared to the control, SSD, and Van groups (*p* < 0.01), indicating an accelerated rate of re-epithelialization. In addition, increased levels of collagen deposition in the PD (Low, high) groups are valid evidence of wound repair. The results of these preclinical evaluations suggest that PD has a high degree of wound repair compared to the control group.

### 2.5. Histological Examination of the Wounds

To determine the mechanism of PD hydrogel efficacy, we performed wound tissue staining for a biomarker of cell proliferation, namely proliferating cell nuclear antigen (PCNA). The recovery of the functional epidermis is highly dependent on cell proliferation and migration. Interestingly, as shown in Figure 5A, the number of PCNA+ cells in the new epidermis and wound bed was significantly higher after PD hydrogel treatment compared to the SSD and Van group. Simultaneously, CD31 expression is a marker of angiogenesis. As displayed in Figure 5B, CD31 expression (mean optical density (MD): 0.29 ± 0.01, 0.43 ± 0.01, respectively) was significantly higher in the PD hydrogel (1%, 2.5%) hydrogel group than in the control group (MD: 0.11 ± 0.03, *p* < 0.001) at day 14, and also higher than in the SSD (MD: 0.15 ± 0.01) and Van groups (MD. 0.24 ± 0.02), which indicated that blood cells were activated during the wound-healing phase (Figure 5B). In general, these results highlight the obvious advantages of PD hydrogels in eliminating bacterial infections and favouring wound healing.

### 2.6. Biosafety Research

Meanwhile, we monitored in vivo toxicity to verify the safety of the hydrogel. Besides, after 14 days of treatment, hematoxylin and eosin staining (H&E) was performed on the major organs of the mice (liver, heart, spleen, lungs, and kidneys) for histopathological changes. There were no histopathological changes compared to the control group (Figure 6). PD did not cause damage or toxicity to the organs during treatment.

### 2.7. Toxicity and Hemocompatibility of PD Hydrogel In Vivo

Moreover, when compared with the blank group, no statistically different body weights were observed for all groups, suggesting the PD almost did not affect their health (Figure 7A). Furthermore, the safety of the hydrogel was evaluated in terms of blood biochemical parameters on day 14, which showed that PD hydrogel did not induce renal or hepatic toxicity as indicated by negligible changes in biomarkers (i.e., alanine transaminase (ALT), aspartate transferase (AST), blood urea nitrogen (BUN) and creatinine (CRE)). Biochemical parameters in the blood showed no significant toxicity in all hydrogels (Figure 7B). As presented in Figure 7C,D, after 1 h of incubation with red blood cells, the PD hydrogel and negative control groups showed no obvious hemolysis, whereas the H2O group showed the bright red colour of hemoglobin. Additionally, the hemolysis ratios of PD (Low, High) hydrogels were 1.1 and 1.32%, respectively, which are below the permissible hemolytic level of 5.0% and thereby indicate the excellent hemocompatibility of the PD hydrogel [21]. All hydrogel groups were deemed to possess positive safety properties. All these findings support that PD can be used as an effective and safe drug for the treatment of MRSA-infected chronic infected wounds.

## 3. Discussion

This study was conducted to investigate whether PD significantly accelerates wound healing in MRSA-infected mice. The results revealed that the process of wound recovery was hampered in untreated MRSA-infected mice. Interestingly, PD hydrogels significantly shorten the wound healing time.

Estimates suggest that, at any given time, millions of people are infected yearly with multidrug-resistant bacteria and the number of deaths associated with such infections is severe. Trdedavancin was approved in 2009 for treating soft tissue skin infections due to Gram-positive bacteria, with the limitation that it has higher nephrotoxicity than vancomycin. Unexpectedly, since the approval of linezolid for clinical use in 2000, drug-resistant strains have been detected in different countries [22]. The rate of resistance to antibiotics by pathogenic microorganisms has far outpaced the development of antibiotics. In particular, bacterial infections may disturb the natural mending process. Chronic, non-healing wounds with drug-resistant strains of bacteria can lead to serious complications, as well as delayed healing. Therefore, research into the development of effective antimicrobial wound dressings represents a growing trend in the wound dressings market.

Wound healing involves three stages: inflammation, proliferation, and remodeling Skin damage is repaired through a coordinated series of responses of hemostasis, inflammation, proliferation, and remodeling, with the risk of non-healing with the dysfunction of any one aspect. The loss of immune and protective mechanisms following skin damage provides a favourable environment for bacterial proliferation. The proliferative phase is usually associated with vascular regeneration, collagen synthesis, granulation tissue formation, epithelialization, and wound contraction [23]. Wound contraction is the main evaluation factor in the healing process of large open wounds. Keratinocytes are the most common cells in the epidermis and form the outermost layer of skin. In the event of injury, keratinocytes migrate from the wound edge to re-epithelialize the damaged tissue and restore the epidermal barrier. All three drugs caused no obvious cytotoxicity at concentrations of 3.9, 7.8, 15.6, 31.25, 62.5 and 125 µg/mL in Figure 1A. Although all three drugs were slightly toxic to cells when the concentration was increased to 125 µg/mL, the viability of human fibroblast HaCaT cells treated with PD was significantly less toxic than that of the SSD and Van groups (Figure 1B). To this point, PD may exhibit a lower toxicity to normal cells at the same concentration and even have a proliferative effect at low concentrations, compared to phloroglucinol and its derivatives [24,25].

In this study, wound-healing tests in vivo were performed in a mouse model to assess the real wound-healing effect of PD hydrogels. Lišková J et al. [20] found that injectable hydrogels containing phloroglucinol inhibited the growth of *E. coli*, without investigating further antimicrobial efficacy in vivo. Previously it was demonstrated that phloroglucinols showed synergistic anti-MRSA activity when paired with doxycycline, but they did not investigate this further as an antibacterial lead [26]. Phloroglucinol derivatives in aspirin BB induced reactive oxygen species to exert their antibacterial activity against S. aureus in vitro [27]. Nevertheless, most reports emphasized the synthesis of phloroglucinol, and do not delve into the antimicrobial mechanisms delve into the antimicrobial mechanism [15,28,29]. How to promote infected wound healing, improve re-epithelialization and reduce bacterial infection is paramount. Thus, analysis of the wound healing including wound area, bacterial load, and yielded encouraging results. Wounds images were captured on days 0, 3, 7, 11, and 14 (Figure 2B). MRSA-infected wounds treated with PD shrank more rapidly than in the control group (Figure 2). Macroscopically, wounds in all groups became smaller with increasing time, and closed faster in PD hydrogel than those in the SSD cream and Van hydrogel groups. The quantification of wound area (Figure 2C–E) showed that infected wounds in the PD hydrogel group healed slightly faster than those in the SSD-cream and Van hydrogel groups. Remarkably, the wounds dressed with high percentage PD hydrogels healed best (6.34% left) on day 14, which were covered with newly formed skin (Figure 2E). It is worth mentioning that the bacterial CFU at the site of infection was obviously decreased on day 14, except for the control group (Figure 3C).

To investigate the effects of PD hydrogel on wound healing, wound tissue was collected and histologically examined using H&E staining. Firstly, epithelialization and granulation tissue formation were analyzed as two important parameters of wound healing. During normal wound healing, an inflammatory response should occur rapidly and last for several days to allow the subsequent stages to occur [30]. Given the increase in inflammation caused by bacterial infections, this reducesthe quality of wound healing. Particularly, MRSA infection can lead to prolongation and exacerbation of the wound, resulting in impaired wound healing [31]. With the healing of the wound sites and re-epidermalization, the number of inflammatory cells decreased [32]. In our research, as illustrated in Figure 4A, the thickness of the regenerated epidermis was thicker in the PD group than in the control, SSD and Van groups, which suggested that the PD group facilitated re-epithelialization during the healing of the wound. More surprisingly, the granulation tissue thickness in both the low PD and high PD groups was much thicker than that in the control, SSD, and Van groups (Figure 4A) suggesting that PD could promote granulation tissue formation.

Local inflammation is fundamental to wound healing and host resistance to infection. Normally, inflammation will slowly disappear over 1 to 2 weeks; it may lengthen and worsen when the tissue is infected [33]. Additionally, Figure 4A revealed that PD treatment significantly reduced the number of inflammatory cells infiltrating the wound edges compared to the other groups, suggesting that PD may have attenuated the inflammatory response, as observed in the macroscopic pictures of the wound. Reducing inflammation may help speed up wound closure. At the same time, the skin consists of up to 70% collagen, which acts as a significant contributor to the skin’s strength, structure and elasticity. Collagen is synthesized during the maturation phase of wound repair and then used as a medium for new tissue, which is closely related to skin strength. Fibroblasts in the dermis initiate collagen synthesis and are responsible for the synthesis, deposition, and remodeling of collagen once it has entered the wound [34]. In this study, it can be seen in Figure 4B, that the PD hydrogel group had a darker blue colour than the control group in wound tissues. Moreover, histochemical analysis (Figure 4A) showed new regenerating epidermal growth at the skin wound. PD groups were almost epithelized in the wound edge and the new epidermis was more continuous and uniform than those analyzed with SSD cream or Van hydrogels and the thickness of the SSD and Van groups was nonuniform and even hypertrophied. When combined with the results in Figure 4, it could be demonstrated that PD gel can be applied for wound repair.

Unlike other reports, our study concentrated more on the role of PD in anti-MRSA and promoting wound repair in the biological mechanism of wound healing. The epidermal function is regenerated by increasing cell proliferation and migration. It was demonstrated that CD31 is involved in angiogenesis during wound healing [35]. Subsequently, the tissue was detected by immunohistochemistry using PCNA as a marker of cell proliferation and CD31 as a marker of angiogenesis [36,37]. The most encouraging result was Figure 5A showed that the number of PCNA-positive keratinocytes per field in the PD group was significantly larger than that in other groups, indicating that the PD could enhance keratinocyte proliferation in vivo and haste the re-epithelialization process. Angiogenesis plays a crucial role in the wound repair process, providing the necessary nutrients and oxygen to the wound site and promoting the formation of granulation tissue [38]. Furthermore, Figure 5B revealed that PD greatly increased the density of CD31-positive vessels compared to the control, SSD, and Van groups. However, no significant differences were found between the SSD and Van groups.

For the practical application of antimicrobial agents, they must have significant antimicrobial power with low toxicity and a good safety profile [39]. Firstly, analysis of paraffin sections of HE-stained mice from major organs by histological studies revealed almost no areas of lesions (Figure 6). HE staining showed that PD did not cause damage or toxicity to the organs. The daily body weights of the mice in each group remained similar throughout the study (Figure 7A). Moreover, blood parameters including AST, ALT, BUN, and CREA were also within normal limits (Figure 7B). Meanwhile, the hemolysis test is illustrated in Figure 7C,D, and shows that the PD hemolysis rate is in the normal range. All these findings indicate clearly that PD can be adapted as an effective and safe platform to eradicate wound infections caused by MRSA bacteria. When compared with commercial SSD cream, the PD hydrogel had better antibacterial activity, reduced inflammation, and promoted angiogenesis, thereby accelerating wound healing. Furthermore, given the potential toxicity of silver accumulation, the safety of PD is better tested when used as a wound dressing for long periods. In summary, the PD hydrogel has biocompatible, hemocompatible and antimicrobial activity and can be used as a safe wound dressing for the treatment of MRSA-induced skin lesions.

There is an urgent need for highly effective antimicrobial agents with no resistance and low toxicity to deal with the overuse of antibiotics and tackle the emergence of superbugs. In this study, we have successfully synthesized PD upfront and now simply encapsulated it in carbomer hydrogel. It is well known that phloroglucinol-based compounds can be used as antibacterial agents against bacteria and fungus. Moreover, carbomer has been extensively adopted as the main drug delivery vehicle for transdermal applications. It has the merit of high viscosity, high compatibility with other drugs, excellent thermal stability and favourable histocompatibility. More importantly, the PD hydrogel dressing is both easy to change and is translucent, allowing for visual viewing of the wound bed without the need to remove the dressing. Of interest, PD performed lower toxicity efficacy in vitro, with cell viability approaching 96.69 ± 5.67% with 125 µg/mL compared to SSD (84.75 ± 17.35%) and Van (96.34 ± 9.34%). Intriguingly, wound recovery in vivo was significantly better in MRSA infections (more than 90% of superbugs were cleared by PD). PD was tested in vivo with an average wound size (18.68% for low-PD and 12.79% for high-PD in MRSA infections). Therefore, our results highlighted that PD could be a new therapeutic strategy and future clinical applications in the management of MRSA infections. However, the standard strains were used without citing more clinical isolates. Accordingly, more clinically isolated strains should be introduced in future studies to further clarify the possibilities of PD in clinical applications. Moreover, the pharmacokinetic profile and therapeutic mechanisms of MRSA still need further detailed study; this may help in developing effective treatment strategies for MRSA-infected wounds.

## 4. Materials and Methods

### 4.1. Materials

Methicillin-resistant *Staphylococcus aureus* MRSA (ATCC 43300) was taken from the Guangdong Microbial Culture Collection Centre (Guangzhou, Guangdong, China). PD was synthesized by ourselves and the structural formula and mass spectra are provided in the Appendix A. Further, in Appendix A, we found that the inhibition circle diameter was larger than that of vancomycin in vitro for the guidelines [40]. Sliver Sulphadiazine (SSD) and the commercialized 1% cream SSD were obtained from Guangdong, China. SSD is a topical antibacterial cream with combined silver and sulfonamide as active ingredients at 1% of the composition. Carbomer 940 (CBM940) is a complex polymer designed as a gelling agent for lipid vesicles in transdermal applications and does not cause skin irritation. Phosphate-buffered saline (PBS, pH = 7.0) was purchased from Gibco-BRL, Los Angeles, CA, USA. Gel carbomer 940 was used as a carrier for PD gel applications. Afterwards, we mixed in two different concentrations of PD, stirred for 5 min, and then adjusted the pH to 7 with triethanolamine. Subsequently, the mixture was filled with ultrapure water (q.s.p 10 mL) to give a 1% Carbopol gel containing 1% (*w*/*v*) and 2.5% (*w*/*v*) PD and stirred periodically at 25 °C for 20 min until a homogeneous mixture was formed, formed as a transparent and well-gelled hydrogel for in vivo wound healing. The vancomycin hydrogel was also produced in the same way. The dose of protective and antimicrobial agents was determined according to the standard dose, and the appropriate pH is around 7.0. SSD cream (1%) and Van hydrogel (1%) were used as drug-positive controls.

### 4.2. Cell Proliferation Assay

The human epidermal keratinocyte (HaCaT) cell line was received from Suyan Biotechnology (Guangzhou, Guangdong, China) and cultured in a DMEM medium mixed with 10% fetal bovine serum, 100 U/mL penicillin and 100 µg/mL streptomycin sulfate. The cells were trypsinized, suspended in media at a concentration of 4 × 10^4^ cells per well, and then plated onto 96-well plates. After 24 h incubation, the medium from each well was replaced with fresh media (100 μL) containing the supernatant from the film–media mixture (each film was put into the complete medium for 24 h) at various concentrations (3.9, 7.8, 15.6, 31.25, 62.25, 125, 250 and 500 µg/mL). After 24 h incubation, a standard MTT viability assay was performed. MTT solution in sterile PBS was mixed to each well and the incubation was continued for 4 h. After that, all solution was then removed from the well and 100 µL DMSO was added to solubilize the crystals. The absorbance measured at 490 nm was comparable to the concentration of viable cells in very well. The untreated cells were used as a control. The data are expressed as mean ± standard deviation.

### 4.3. Cell Apoptosis Assay by Hoechst/Propidium Iodide (PI) Staining

Furthermore, to observe the cell morphology, HaCaT cells were cultured to 80% and treated with 0.8 mM, of three types of medicine (PD, SSD, Van) for 24 h, respectively. After washing 3 times with phosphate-buffered saline (PBS), the cells were added to 10 mL Hoechst 33,342 (5 µg/mL) and incubated at 37 °C for 10 min, followed by adding PI (10 µg/mL) at the darkroom temperature for 10 min. Fluorescence images of HaCaT cells were obtained under a fluorescence microscope (Olympus, Tokyo, Japan).

### 4.4. In Vivo Antibacterial Activity Test

All animal procedures were performed following the Guidelines for Care and Use of Laboratory Animals of Guangdong Pharmaceutical University and approved (NO. gdpulacspf2017582) by the Animal Ethics Committee according to the principles outlined in the Declaration for all animal experimental investigations. Six-week female Balb/c mice (18 ± 2 g) were obtained from China. All mice were maintained on a 12-h light/dark cycle in a room at 22–25 °C and allowed free access to food and water before surgery. All mice were anaesthetized with isoflurane and circular wounds (diameter of 10 mm) were created on the backbones of the mice. Thereafter, 100 μL of MRSA (1 × 10^8^ CFU mL^−1^) was smeared on the surface of the wounds to establish the MRSA-infected mouse model. One day after continuous infection, all mice were randomly divided into four groups. The hydrogel was applied directly to the backs of mice wound formed on the back of the mouse twice a day with continuous application for 14 days. The mice in these different groups were photographed and the size of infected wounds was measured every day to evaluate the healing efficacy. After treatment for 14 days, all mice were sacrificed, and the tissues of infected wound areas and major organs were harvested for pathological histology analysis.

### 4.5. Bacterial Load

Wound healing is a complex process characterized by collagen synthesis, re-epithelialization, tissue synthesis, and extracellular matrix remodeling [41]. Skin tissues from animals with MRSA wound infections taken from different groups of animals were soaked in phosphate-buffered saline and cultured on agar plates to estimate the bacterial population.

### 4.6. Histopathological Examination

The mice were sacrificed on day eight post-surgery, and then the wound tissues with adjacent normal skin (10 × 10 mm) were carefully harvested and fixed in 4% paraformaldehyde for 24 h. Next, the tissues were embedded in paraffin and sliced at a thickness of 5 μm. H&E staining was performed for histological analysis. Each of the tissue samples was then subjected to Masson’s trichrome (MT) staining. The number of inflammatory cells in the wound edge, the thickness of granulation tissue, and length of the regenerated epidermis, which was defined as the distance from the border between normal skin and wound region to the anterior edges of the newly generated epidermis, were quantified using ImageJ software by two independent researchers.

### 4.7. Immunohistochemistry Analysis

Immunohistochemistry analysis was conducted as previously described [42]. Briefly, wound tissue sections were deparaffinized, rehydrated, and boiled in a 100 °C citrate buffer bath. Next, the sections were treated with primary antibodies at 4 °C overnight. The primary antibodies obtained from Abcam (UK) were as follows: anti-CD31 (c) antibody at 1:250 dilution, ab28364; anti-PCNA (proliferating cell nuclear antigen) antibody at 1:200 dilution, ab15497. After incubation with goat-anti-rabbit IgG antibody and avidin–peroxidase reagent, the sections were stained with 3, 30-diaminobenzidine tetrahydrochloride (DAB) solution and hematoxylin and then were photographed under a microscope.

### 4.8. Biocompatibility Test

#### 4.8.1. Biological Safety Study

Fasting body masses were recorded on days 0, 3, 7, 11, and 14 of the administration of the drug to mice using a small animal weighing scale. On the 14th day, all mice were sacrificed, and the tissues of infected wound areas and major organs were harvested for pathological histology analysis. In the in vivo wound healing assay, blood samples were obtained from treated MRSA wound-infected mice and then used for biochemical analysis.

#### 4.8.2. Hemolysis Assay

The hemolysis activity assay was performed according to reference [21]. To obtain the erythrocytes, the mice’s blood was centrifuged (at 1000 rpm) for 10 min. It was washed three times with Tris buffer and then diluted to a final concentration of 5% (*v*/*v*). Hydrogel (500 µL) was mixed with the obtained erythrocytes (500 µL) into a 24-well microplate and incubated statically at 37 °C for 1 h with a shaking speed of 150 rpm. Following that, the microplate contents were centrifuged (at 1000 rpm) for 10 min and the supernatant (100 µL) was pipetted into a 96-well microtiter plate. H2O was included as a positive control and PBS as a negative control. The automated microplate reader (Sunrise, Tecan, Switzerland) was utilized to detect the absorbance at 570 nm. The formula of hemolysis percentage: Hemolysis (%) = [(AX − Ab)/(At − Ab)] × 100% where AX was the absorbance value for the (hydrogel) experiment group. At and Ab represents the absorbance value of H2O and PBS, respectively.

### 4.9. Statistical Analyses

Data are shown as mean ± standard deviations (SD) in the graphs. Data analysis was performed using SPSS statistical software (version 26.0). Graphs were constructed using GraphPad Prism 8.0 (GraphPad Software, San Diego, CA, USA). Statistical significance was determined by a one-way ANOVA followed by the LSD post hoc test. *p* < 0.05 were acknowledged as statistically significant.

## Figures and Tables

**Figure 1 ijms-23-08682-f001:**
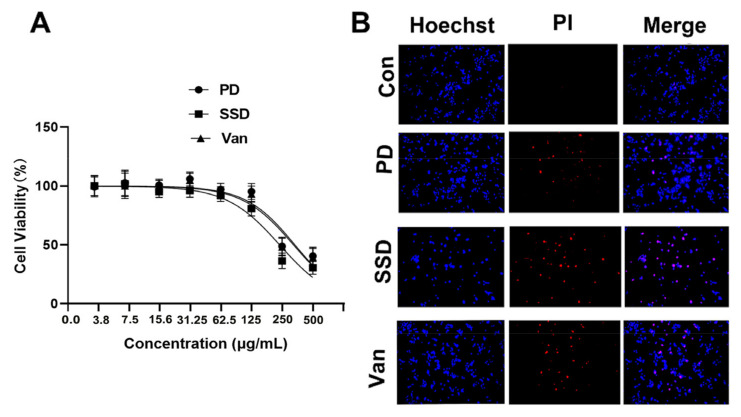
The cytotoxicity of the PD, SSD, and Van on human keratinocytes (HaCaT). (**A**) HaCaT cells were treated with the indicated concentration of the PD, SSD, and Van for 24 h. Cell viability was assessed by MTT assay. (**B**) HaCaT cells were treated with 125 µg/mL of the PD, SSD, and Van for 24 h. Cell apoptosis was assessed by Hoechst/PI staining. The scale bar is 200 μm.

**Figure 2 ijms-23-08682-f002:**
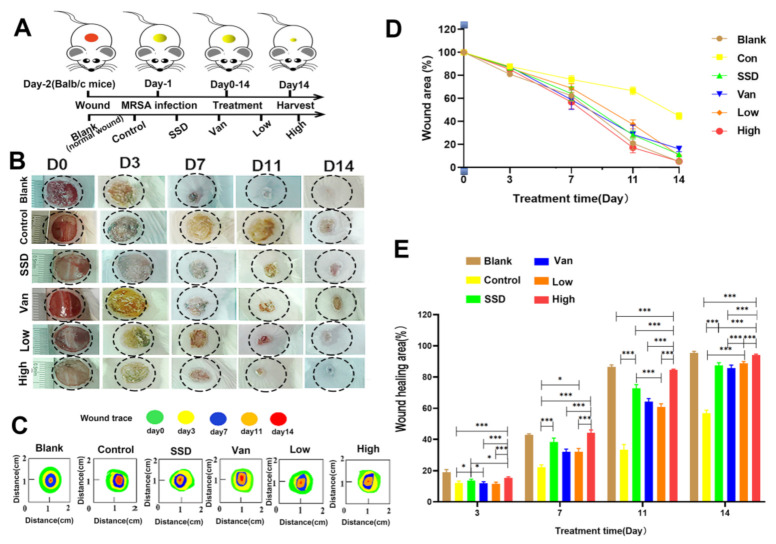
In vivo effects of the composite membranes on wound healing. (**A**) Schematic illustration of the construction and treatment process of wound infection. (**B**)The representative images of the wound healing process of mice treated with blank, control, PD (Low, High), SSD, and Van hydrogels. (**C**)Traces of wound-bed closure during 14 days for each treatment. (**D**) In vivo wound closure rates for six groups at different time points. (**E**) Quantitation of the topical wound healing rate. The data are expressed as mean ± standard deviation and analyzed using a one-way ANOVA followed by the LSD test for multiple comparisons (* *p* < 0.05, *** *p* < 0.001, *n* = 6).

**Figure 3 ijms-23-08682-f003:**
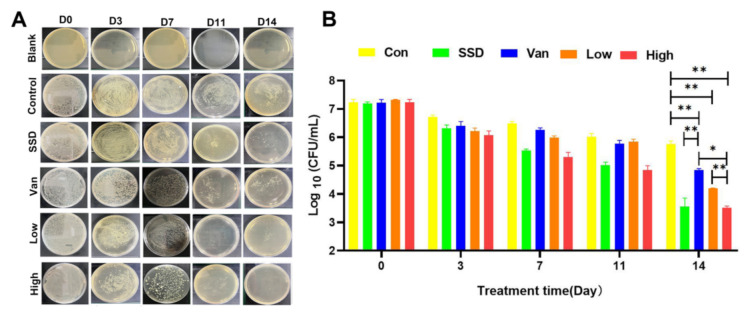
(**A**) The photographs of bacterial colonies spread on agar plates from the infected skin tissues with different treatments. Representative images of bacterial growth on agar plates after 24 h of incubation. (**B**) Corresponding statistical analysis of the viable MRSA. Bacterial load (log CFU/mL) in various formulation-treated groups as compared to untreated control. The data are expressed as mean ± standard deviation and analyzed using a one-way ANOVA followed by the LSD multiple comparison tests (* *p* < 0.05, ** *p* < 0.01, *n* = 3).

**Figure 4 ijms-23-08682-f004:**
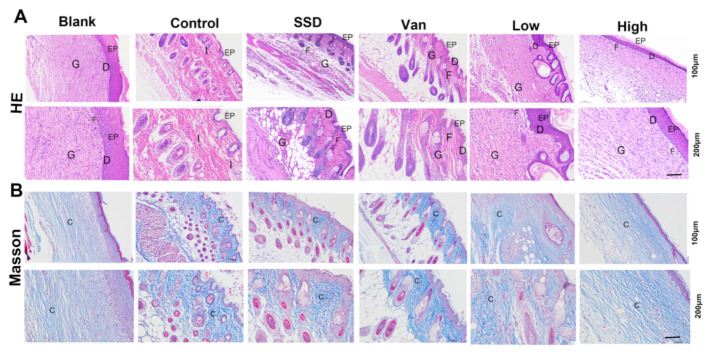
(**A**) Typical H&E staining micrographs of the wounds treated by different treatments on day 14. (**B**) Representative images of Masson trichrome-stained tissue sections of wounds treated with PD (1%, 2.5%), SSD, and Van wounds on day 14. Notice the dense collagen deposition in the PD, SSD, and Van nanofibrous dressings compared to the untreated wound control (magnifications, ×100 and ×200). EP: epidermis; D: dermis; G; granulation tissue; I: inflammatory cells; F: fibroblasts; C: collagen.

**Figure 5 ijms-23-08682-f005:**
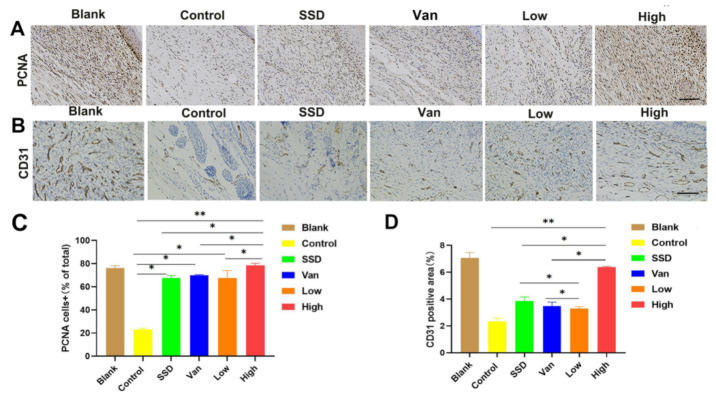
Immunohistochemical analysis of cutaneous wound sections at day 14 post-wound healing/infection. (**A**) Analysis of proliferating cell nuclear antigen (PCNA) + cells in wounds at day 14 post-infection. Representative images of PCNA+ cells in wounds treated with Pd, SSD, Van, or the blank hydrogel. (**B**) Analysis of CD31 in wounds at day 14 post-infection. (**C**) Corresponding quantification of PCNA+ cells expressed as a percentage of total cell counts. (**D**) Statistical analysis of CD31 expressed at the wound site in different treatment groups on day 14. All data are presented as mean ± standard deviation. Significant differences are indicated with asterisks (* *p* < 0.05, ** *p* < 0.01, *n* = 3. (magnifications, ×200).

**Figure 6 ijms-23-08682-f006:**
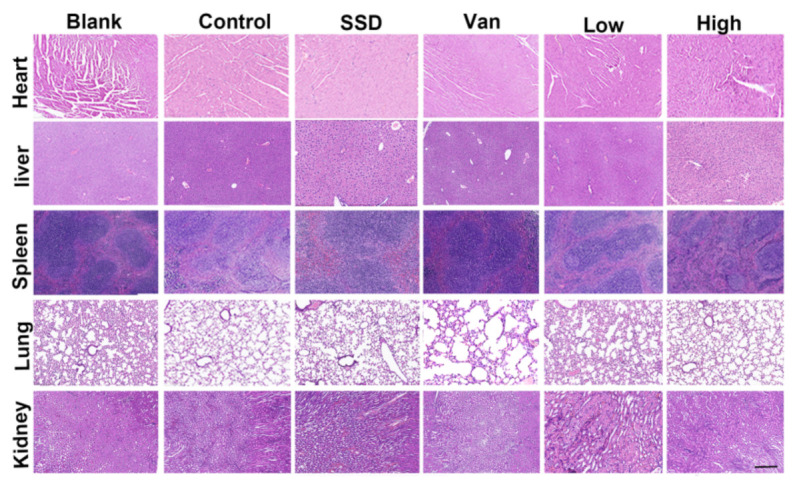
H&E staining of the heart, liver, spleen, lung, and kidney on day 14 after different treatments (magnifications, ×200).

**Figure 7 ijms-23-08682-f007:**
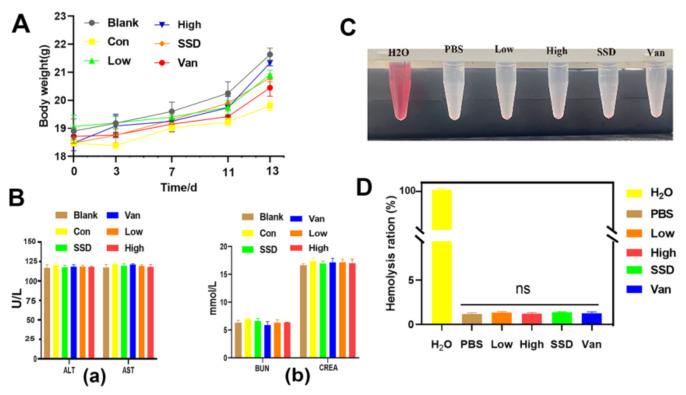
Biosafety assay. (**A**) Body weight assessment of MRSA skin-infected mice by various therapies. (**B**) Liver and kidney function was assessed by measuring (a) ALT, alanine transferase; AST, aspartate transferase; (b) BUN, blood urea nitrogen; CREA, creatinine. (**C**) Photographs from the hemolytic assay. (**D**) Quantitative data of the hemolytic ratio, with the data shown as means ± standard deviation. “ns” indicates no significance (*p* > 0.05, *n* = 3).

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
