# Peer review of "Phloroglucinol Derivative Carbomer Hydrogel Accelerates MRSA-Infected Wounds’ Healing"

_ijms, 2022, doi:10.3390/ijms23158682_

Round 1
Reviewer 1 Report
Author Huang et al. describes "Phloroglucinol Derivative Carbomer Hydrogel Accelerates MRSA-infected Wounds' Healing".
This paper can be accepted after considering the following comments;
1: INtroduction needs to be added to the multiple application forms of the phloroglucinol by citing the latest references;
- DOI: 10.1016/j.colsurfb.2021.112307
2: Add a reference related to line 122 DOI: 10.3390/md20060384
3: How the cytotoxicity effect of this derivative is compared to the previously reported cytotoxicity effect of phloroglucinol and its derivatives.
4: Figure 6 legend is missing
5: Try to compare more about your finding with the previous finding in the discussion sections.
Author Response
Response to Reviewer 1 Comments
Dear Editors and Reviewers:
Thank you again for the reviewers’ comments concerning our manuscript entitled “Phloroglucinol Derivative Carbomer Hydrogel Accelerates MRSA-infected Wounds' Healing” (ID: ijms-1845224). These comments are all valuable and very helpful for revising and improving our paper quality, as well as the important guiding significance to our research. We have studied these comments and made the corresponding revisions carefully. The main corrections in the paper and the responses to the reviewer’s comments are as follows:
Notes:
For references, we have deleted some literature that is not relevant to the content of the manuscript.
For the extensive English revisions in this manuscript, we have checked by a native English-speaking colleague.
To make it clearer to reviewers, we have used red font in the paper to indicate the main changes we made. Meanwhile, revisions in this manuscript were marked up using the “Track Changes” function due to using MS Word/LaTeX.
Reviewer 1
Comments and Suggestions for Authors
Author Huang et al. describes "Phloroglucinol Derivative Carbomer Hydrogel Accelerates MRSA-infected Wounds' Healing".
This paper can be accepted after considering the following comments;
Point 1: Introduction needs to be added to the multiple application forms of the phloroglucinol by citing the latest references;
DOI: 10.1016/j.colsurfb.2021.112307
Response 1: Thank you for your valuable proposal. We have added the information about the multiple application forms of phloroglucinol on page 3 presented in lines 107 -114 of < 1. Introduction >.
"Phloroglucinol encapsulated in chitosan nanoparticles by ionic gel technology is more effective against bacteria such as Staphylococcus aureus than pure phloroglucinol [18]. The addition of the polyphenols phloroglucinol to chitosan hydrogels reduced the growth of E. coli and also improved the antioxidant activity of the hydrogels [19]. Several studies have discussed the promising future of phloroglucinol and its derivatives in conjugation, nanoformulations, antibiotic combinations and encapsulation in antibacterial applications [20]. "
References
- Javed A., Hussain MB., Tahir A., Waheed M., Anwar A., Shariati MA., Plygun S., Laishevtcev A., Pasalar M. Pharmacological Applications of Phlorotannins: A Comprehensive Review. Curr Drug Discov Technol 2021; 18: 282-292. doi: 10.2174/1570163817666200206110243
- Khan F., Oh D., Chandika P., Jo DM., Bamunarachchi NI., Jung WK., Kim YM. Inhibitory activities of phloroglucinol-chitosan nanoparticles on mono- and dual-species biofilms of Candida albicans and bacteria. Colloids Surf B Biointerfaces 2022; 211:112307. doi: 10.1016/j.colsurfb.2021.112307.
20.Lišková J, Douglas TE, Beranová J, Skwarczyńska A, Božič M, Samal SK, Modrzejewska Z, Gorgieva S, Kokol V, Bačáková L. Chitosan hydrogels enriched with polyphenols: Antibacterial activity, cell adhesion and growth and mineralization. Carbo-hydr Polym. 2015; 29:135-42. doi: 10.1016/j.carbpol.2015.04.043.
Point 2: Add a reference related to line 122. DOI: 10.3390/md20060384.
Response 2: Thank you for your valuable proposal. We have added the reference in the text at the corresponding place. We have added the information mentioned above on page 3 presented in line 107 of < 1. Introduction >.
"It was reported that phlorotannins possess biological activities including antimicrobial, antidiabetic, anticancer, antiviral, and antioxidant [17]."
Point 3: How the cytotoxicity effect of this derivative is compared to the previously reported cytotoxicity effect of phloroglucinol and its derivatives?
Response 3: Thank you for your valuable comment. This section indeed needs some clarification.
Firstly, phloroglucinol at 0.3 mM no marked cytotoxicity was noted in a normal colon cell line (Lopes-Costa E et al., 2017). Quéguineur B et al. (2012) demonstrated that phloroglucinol (0.004-0.4 mM) had no impact on cell viability (human HepG2 cell line). In addition, phloroglucinol derivative inhibited the proliferation of MCF-7 cells with IC50 esteem at 76.10 μM (Zhang Y et al., 2012). Phloroglucinol derivative (0.1, 1, and 10 µg/mL) had a non-toxic effect in HDFs and HUVECs cells (Moghadam SE et al., 2019). Based on the mentioned concentrations and pre-experiments, we determined the concentrations of PD at 3.8, 7.5, 15.6, 31.25, 62.5, 125, 250 and 500 µg/mL change anout 0.01, 0.02, 0.05, 0.10, 0.21, 0.42, 0.85, and 1.7 mM in our study. Furthermore, the MTT assay showed no obvious cytotoxicity at concentrations of PD ( 3.8, 7.5, 15.6, 31.25, 62.5 and 125 µg/mL) in HaCaT cells. Notably, at 250 µg/mL of PD with cell survival rate was 54.34 ± 5.67 %. Thus, compared with the previously reported cytotoxicity effect of phloroglucinol and its derivatives, this phloroglucinol derivative may exhibit lower toxicity to normal cells at the same concentration and even have a proliferative effect at low concentrations.
We have added the information about the multiple application forms of phloroglucinol on page 9 presented in lines 308-310 of < 1. Introduction >.
"To this point, PD may exhibit lower toxicity to normal cells at the same concentration and even have a proliferative effect at low concentrations, compared to phloroglucinol and its derivatives [24, 25]. "
References
- Lopes-Costa E, Abreu M, Gargiulo D, Rocha E, Ramos AA. Anticancer effects of seaweed compound fucoxanthin and phloroglucinol, alone and in combination with 5-fluorouracil in colon cells. J Toxicol Environ Health A. 2017; 80: 776-787. DOI: 10.1080/15287394.2017.1357297.
- Quéguineur B, Goya L, Ramos S, Martín MA, Mateos R, Bravo L. Phloroglucinol: antioxidant properties and effects on cellular oxidative markers in human HepG2 cell line. Food Chem Toxicol. 2012; 50: 2886-93. DOI: 10.1016/j.fct.2012.05.026.
- Zhang Y, Luo M, Zu Y, Fu Y, Gu C, Wang W, Yao L, Efferth T. Dryofragin, a phloroglucinol derivative, induces apoptosis in human breast cancer MCF-7 cells through ROS-mediated mitochondrial pathway. Chem Biol Interact. 2012; 199: 129-36. DOI: 10.1016/j.cbi.2012.06.007.
- Moghadam SE, Moridi Farimani M, Soroury S, Ebrahimi SN, Jabbarzadeh E. Hypermongone C Accelerates Wound Healing through the Modulation of Inflammatory Factors and Promotion of Fibroblast Migration. Molecules. 2019; 24: 2022. DOI: 10.3390/molecules24102022.
Point 4: Figure 6 legend is missing.
Response 4: We thank the reviewer for this comment. I'm sorry that I didn't note them in time due to my negligence. We have added the legend on page 8 presented in lines 251-252 <2. Results section 2.6 Biosafety research>.
"Figure.6 H&E staining of the heart, liver, spleen, lung, and kidney on day 14 after different treatments (magnifications, ×200). "
Point 5: Try to compare more about your finding with the previous finding in the discussion sections.
Response 5: This comment was highly appreciated. This comment was highly appreciated and provided us with a very good suggestion. We have added the comparison of our findings with previous findings on page 9 presented in lines 312-320 < 3. Discussion>.
"Although Lišková J et al. [20] found that injectable hydrogels containing phloroglucinol inhibited the growth of E. coli, without investigating further antimicrobial efficacy in vivo. Previously it was demonstrated that phloroglucinols showed synergistic anti-MRSA activity when paired with doxycycline, but did not investigate further as an antibacterial lead [26]. Phloroglucinol derivatives in aspirin BB induced reactive oxygen species to exert their antibacterial activity against S. aureus in vitro [27]. Nevertheless, most reports emphasized the synthesis of phloroglucinol, and do not delve into the antimicrobial mechanisms delve into the antimicrobial mechanism [28, 29, 30]. "
References
- Lišková J, Douglas TE, Beranová J, Skwarczyńska A, Božič M, Samal SK, Modrzejewska Z, Gorgieva S, Kokol V, Bačáková L. Chitosan hydrogels enriched with polyphenols: Antibacterial activity, cell adhesion and growth and mineralization. Carbohydr Polym. 2015; 20; 135-42. DOI: 10.1016/j.carbpol.2015.04.043.
26.Mittal N., Tesfu HH., Hogan AM., Cardona ST., Sorensen JL. Synthesis and antibiotic activity of novel acylated phloroglucinol compounds against methicillin-resistant Staphylococcus aureus. J Antibiot (Tokyo) 2019; 72: 253-259. DOI: 10.1038/s41429-019-0153-4.
27.Li N., Gao C., Peng X., Wang W., Luo M., Fu YJ., Zu YG. Aspidin BB, a phloroglucinol derivative, exerts its antibacterial activity against Staphylococcus aureus by inducing the generation of reactive oxygen species. Res Microbiol 2014; 165: 263-72. DOI: 10.1016/j.resmic.2014.03.002.
28.Khan N., Rasool S., Ali Khan S., Bahadar Khan S. A new antibacterial dibenzofuran-type phloroglucinol from myrtuscommun-islinn. Nat Prod Res 2020; 34:3199-3204. DOI: 10.1080/14786419.2018.1556657.
29.Mo QH., Yan MQ., Zhou XL., Luo Q., Huang XS., Liang CQ. Phloroglucinol derivatives rhotomensones A-G from Rhodomyrtus tomentosa. Phytochemistry 2021; 190: 112890. DOI: 10.1016/j.phytochem.2021.112890.
30.Julian WT., Vasilchenko AV., Shpindyuk DD., Poshvina DV., Vasilchenko AS. Bacterial-Derived Plant Protection Metabolite 2,4-Diacetylphloroglucinol: Effects on Bacterial Cells at Inhibitory and Subinhibitory Concentrations. Biomolecules 2020; 11: 13. doi: 10.3390/biom11010013.
Here, we are extremely appreciative of your friendly advices and suggestion from the bottom of our hearts. The review process of this manuscript means a lot to us, and it is valuable to our later research. We really appreciate the Editor and Reviewers' comments and suggestions to us, which improved the quality of this manuscript. We look forward to hearing from you regarding our submission. We would be glad to respond to any further questions and comments that you may have.
Thanks earnestly.

Reviewer 2 Report
The article Phloroglucinol derivative carbomer hydrogel accelerates MRSA-infected wounds' healing is suitable for publication in Int. J. Mol. sci. mainly due to the relevance of the development of effective drugs with antibacterial properties for use as a wound healing dressing.
The manuscript can be accepted for publication in IJMS after the following comments have been minor revision:
- The abstract is too long and should be more concise while still retaining the most important information.
- The purpose of the study should be formulated more clearly, justify the scientific and practical purpose of your study.
- What are phloroglucinol derivatives? In materials and methods, authors should indicate whether they synthesized PD themselves or purchased a commercially available product.
- The viability of HaCaT cells treated with 125 μg/mL PD was 96.69 ± 5.67%, and what is the viability of cells at a concentration of 250 μg/mL?
- Figures 2A, 3A, 5C and D are disproportionately elongated. Get your drawings right.
- In Fig. 4 are missing letters.
- In section 2.5. there is no description of immunohistochemical analysis for CD31 antigens.
- There is no title for Figure 6.
- Line 269, what kind of figure is this 7A?
- Conclusions should be expanded and contain the main results in quantitative reports.
Author Response
Response to Reviewer 2 Comments
Dear Editors and Reviewers:
Thank you again for the reviewers’ comments concerning our manuscript entitled “Phloroglucinol Derivative Carbomer Hydrogel Accelerates MRSA-infected Wounds' Healing” (ID: ijms-1845224). These comments are all valuable and very helpful for revising and improving our paper quality, as well as the important guiding significance to our research. We have studied these comments and made the corresponding revisions carefully. The main corrections in the paper and the responses to the reviewer’s comments are as follows:
Noted:
For figures, replaced the images on Figures 2A, 3A, 5C and D with new ones on Figures 2A, 3A, 5C and D to answer the question of Reviewer 2.
Supplemented the figure letters in Figure 4 to answer the question of Reviewer 2.
For references, we have deleted some literature that is not relevant to the content of the manuscript.
For the extensive English revisions in this manuscript, we have checked by a native English-speaking colleague.
To make it clearer to reviewers, we have used red font in the paper to indicate the main changes we made. Meanwhile, revisions in this manuscript were marked up using the “Track Changes” function due to we using MS Word/LaTeX.
Reviewer 2
Comments and Suggestions for Authors
The article Phloroglucinol derivative carbomer hydrogel accelerates MRSA-infected wounds' healing is suitable for publication in Int. J. Mol. sci. mainly due to the relevance of the development of effective drugs with antibacterial properties for use as a wound healing dressing.
The manuscript can be accepted for publication in IJMS after the following comments have been minor revisions:
Point 1: The abstract is too long and should be more concise while still retaining the most important information.
Response 1: Thank you for your valuable comment. We have condensed the summary and kept its most important information in lines 17-32 of < Abstract> on page 1.
" Abstract: Globally, wound infection is considered to be one of the major healthcare problems, with bacterial infections being the most critical threat, leading to poor and delayed wound healing, and even death. As a superbug, Methicillin-resistant Staphylococcus aureus (MRSA) causes a profound hazard to public health safety, prompting us to search for alternative treatment approaches. Herein, the MTT test and Hoechst/propidium iodide (PI) staining demonstrated that PD was slightly less toxic to human fibroblasts including Human keratinocytes (HaCaT) cell line than Silver sulfadiazine (SSD), and Vancomycin (Van). In the MRSA-infected wound model, PD hydrogel (1 %, 2.5 %) was applied with for 14 days. The wound healing of PD hydrogel groups was superior to the SSD, Van, and control groups. Remarkably, the experimental results showed that PD reduced the number of skin bacteria, reduced inflammation, and upregulated the expression of PCNA (keratinocyte proliferation marker), and CD31 (angiogenesis manufacturer) at the wound site by histological (including hematoxylin-eosin (HE) staining, Masson staining) and immunohistochemistry. Additionally, no toxicity and haemocompatibility or histopathological changes to organs were observed. Altogether, these results suggested the potential of PD hydrogel as a safe, effective, and low toxicity hydrogel for the future clinical treatment of MRSA-infected wounds. "
Point 2: The purpose of the study should be formulated more clearly, and justify the scientific and practical purpose of your study.
Response 2: This comment shows us very precise scientism. We have formulated clearly the purpose of our study in lines 116-121 of < Introduction> on page 3.
"Given the critical situation caused by drug-resistant bacteria and the many reports finding good antibacterial effects of phloroglucinol, besiedes, achieving the goal of developing new treatments and reducing the medical burden associated with MRSA-infected skin wounds. Therefore, we synthesised PD and we aimed to investigate whether it has low toxicity to cell growth in vitro and better anti-MRSA wound infection in vivo."
Point 3: What are phloroglucinol derivatives? In materials and methods, authors should indicate whether they synthesized PD themselves or purchased a commercially available product.
Response 3: We thank the reviewer for this comment. I'm sorry that I didn't note them in time due to my negligence. This point indeed needs some clarification.
Need to be clear that we synthesized PD ourselves. More importantly, we have synthesized more than 20 derivatives of phloroglucinol and screened them for antibacterial activity and found that one of them has strong anti-MRSA activity. Notably, the synthesis of the drug is in another journal submission from our group and we can only provide the structural formula and mass spectra of the compound in the supplementary material. Moreover, we found that the in vitro antibacterial activity of PD was stronger than that of vancomycin, and the diameter of its inhibition circle was larger than that of vancomycin. Similarly, images of our inhibition circles are presented in the supplementary materials.
We have supplemented the PD information in lines 424-428 of <4. Materials and methods section 4.1 Materials > on page 12.
" PD was synthesized by ourselves and the structural formula and mass spectra were provided in the supplementary material (Figure S1). Further, in Fig. S2, we found that the inhibition circle diameter was larger than that of vancomycin in vitro for the guidelines [41]. "
Point 4: The viability of HaCaT cells treated with 125 μg/mL PD was 96.69 ± 5.67%, and what is the viability of cells at a concentration of 250 μg/mL?
Response 4: I'm sorry that I didn't note it in time due to my negligence.
We have added the cell viability of HaCaT cells treated with 250 μg/mL) PD on page 3 presented in lines 127-129 of <MATERIALS AND METHODS section 2.2 MTT aasay>.
"Notably, at 250 μg/mL of SSD, and Van, most of the cells died and floated, with cell survival rates only 32.56 ± 4.05 % and 40.68 ± 6.34 %, respectively, as compared to 54.34 ± 5.67 % under PD treatment."
Point 5: Figures 2A, 3A, 5C and D are disproportionately elongated. Get your drawings right.
Response 5: We are very sorry for our image processing. We have corrected the proportionately of Figures 2A, 3A, 5C and D.
Point 6: In Fig. 4 are missing letters.
Response 6: We feel very sorry that we didn't add it due to our negligence. The letters were supplemented in Fig.4.
Point 7: In section 2.5. there is no description of immunohistochemical analysis for CD31 antigens.
Response 7: We feel very sorry that we didn't add it in time due to our negligence.
We have added the analysis for CD31 antigens in lines 226-231 of <Results section> on page 7.
"Simultaneously, CD31 expression is a marker of angiogenesis. As displayed in Fig. 5B, CD31 expression (mean optical density (MD): 0.29 ± 0.01, 0.43 ± 0.01, respectively) was significantly higher in the PD hydrogel (1 %, 2.5 %) hydrogel group than in the control group (MD: 0.11 ± 0.03, P<0.001) at day 14, and also higher than in the SSD (MD: 0.15 ± 0.01) and Van groups (MD. 0.24 ± 0.02), which indicated that blood cells were activated during the wound healing phase (Figure 5B). "
Point 8: There is no title for Figure 6.
Response 8: We feel very sorry for our negligence in missing the title for Figure 6. The specific information was added in lines 244-249 of <Results section> on page 7.
2.6 Biosafety research
Meanwhile, we monitored in vivo toxicity to verify the safety of the hydrogel. Besides, after 14 days of treatment, hematoxylin and eosin staining (H&E) on the major organs of the mice (liver, heart, spleen, lungs, and kidneys) for histopathological changes. There were no histopathological changes compared to the control group (Fig. 6). PD did not cause damage or toxicity to organs during treatment.
Point 9: Line 269, what kind of figure is this 7A?
Response 9: We feel very sorry for our negligence in missing the figure.7.
Figure.7A means the low toxicity of PD to the body was confirmed with negligible changes in body weight in mice compared to the blank, control, and treatment groups. The information was supplemented in lines 274-276 of <Results section> on page 8.
"Moreover, compared with the blank group, no statistically different body weights were observed for all groups, suggesting the PD almost did not affect their health (Fig. 7A). "
Point 10: Conclusions should be expanded and contain the main results in quantitative reports.
Response 10: Thanks to you for your good comments. This comment is highly appreciated and provide us with a very good suggestion.
The specific information has been added in lines 398-419 of < Discussion section> on page 11.
" The urgent need for highly effective antimicrobial agents with no resistance and low toxicity to deal with the overuse of antibiotics and tackle the emergence of superbugs. In this study, we have successfully synthesized PD upfront and now simply encapsulate it in carbomer hydrogel. It is well known that phloroglucinol-based compounds can be used as antibacterial agents against bacteria and fungus. Moreover, carbomer has been extensively adopted as the main drug delivery vehicle for transdermal applications. It has the merit of high viscosity, high compatibility with other drugs, excellent thermal stability and favourable histocompatibility. More importantly, the PD hydrogel dressing is both easy to change and translucent, allowing visual viewing of the wound bed without the need to remove the dressing. Of interest, PD performed lower toxicity efficacy in vitro, with cell viability approaching 96.69±5.67 % with 125 μg/mL compared to SSD (84.75 ± 17.35 %) and Van(96.34 ± 9.34 %). Intriguingly, wound recovery in vivo was significantly better in MRSA infections (more than 90 % of superbugs were cleared by PD). PD was tested in vivo with average wound size (18.68 % for low-PD and 12.79 % for high-PD in MRSA infections). Therefore, our results highlighted that PD could be a new therapeutic strategy and future clinical applications in the management of MRSA infections. However, the standard strains were used without citing more clinical isolates. Accordingly, more clinical isolated strains should be introduced in future studies to further clarify the possibilities of PD in clinical applications. Moreover, the pharmacokinetic profile and therapeutic mechanisms of MRSA still need further detailed study, this may help in developing effective treatment strategies for MRSA-infected wounds. "
Here, we are extremely appreciative of your friendly advices and suggestion from the bottom of our hearts. The review process of this manuscript means a lot to us, and it is valuable to our later research. We really appreciate the Editor and Reviewers' comments and suggestions to us, which improved the quality of this manuscript. We look forward to hearing from you regarding our submission. We would be glad to respond to any further questions and comments that you may have.
Thanks earnestly.
